# The Effect of Titanium Surface Topography on Adherent Macrophage Integrin and Cytokine Expression

**DOI:** 10.3390/jfb14040211

**Published:** 2023-04-11

**Authors:** Manju Sofia Pitchai, Deepak Samuel Ipe, Stephen Hamlet

**Affiliations:** School of Medicine and Dentistry, Gold Coast Campus, Griffith University, Southport, QLD 4222, Australia

**Keywords:** macrophage, integrin, cytokine, polarization, titanium, surface, topography

## Abstract

Immunomodulatory biomaterials have the potential to stimulate an immune response able to promote constructive and functional tissue remodeling, as opposed to persistent inflammation and scar tissue formation. This study examined the effects of titanium surface modification on integrin expression and concurrent cytokine secretion by adherent macrophages in vitro in an attempt to delineate the molecular events involved in biomaterial-mediated immunomodulation. Non-polarised (M0) and inflammatory polarised (M1) macrophages were cultured on a relatively smooth (machined) titanium surface and two proprietary modified rough titanium surfaces (blasted and fluoride-modified) for 24 h. The physiochemical characteristics of the titanium surfaces were assessed by microscopy and profilometry, while macrophage integrin expression and cytokine secretion were determined using PCR and ELISA, respectively. After 24 h adhesion onto titanium, integrin α1 expression was downregulated in both M0 and M1 cells on all titanium surfaces. Expression of integrins α2, αM, β1 and β2 increased in M0 cells cultured on the machined surface only, whereas in M1 cells, expression of integrins α2, αM and β1 all increased with culture on both the machined and rough titanium surfaces. These results correlated with a cytokine secretory response whereby levels of IL-1β, IL-31 and TNF-α increased significantly in M1 cells cultured on the titanium surfaces. These results show that adherent inflammatory macrophages interact with titanium in a surface-dependent manner such that increased levels of inflammatory cytokines IL-1β, TNF-α and IL-31 secreted by M1 cells were associated with higher expression of integrins α2, αM and β1.

## 1. Introduction

Biomaterial-mediated activation of macrophages and modulation of their phenotypes have emerged as key strategies to improve the efficacy with which the biomaterial enhances tissue integration and repair, as opposed to a foreign-body response characterized by fibrous encapsulation and biomaterial isolation [1,2]. However, to elicit predictable immune responses to biomaterials such as titanium, there is a need for a thorough understanding of how the topographical and physiochemical properties of the biomaterial can affect the function of immunological mediators such as macrophages following their attachment to its surface.

Following implantation, biomaterials are immediately coated with proteins such as fibronectin, vitronectin, albumin, complement, etc., that are adsorbed onto the surface. Activated platelets subsequently release chemoattractants that direct the migration of macrophages to the wound site, where they bind to the biomaterial surface via integrin-mediated interactions with the adsorbed proteins [3,4]. The subsequent activation and modulation of macrophage function is therefore a key early element in the overall wound healing process following biomaterial implantation.

Macrophages are an essential component of innate immunity and play a central role in inflammation and host defense. Moreover, these cells fulfill homeostatic functions beyond defense, including tissue remodeling in ontogenesis and orchestration of metabolic functions [5]. At the cell-matrix interface, the mechanical interaction(s) between macrophage cells and adsorbed protein occur through focal adhesions that link extracellular matrix proteins to the cellular contractile cytoskeleton. Trans-membrane integrins thus serve as linker proteins that facilitate focal adhesion by connecting the extracellular substrates to actin stress fibers extended from within cells [6]. Subsequent cytoskeleton rearrangements have been demonstrated to be able to activate a wide range of important intracellular signaling pathways such as PI3 K/Akt, MAPK and FAK, which in turn can regulate a wide range of cellular responses including growth, differentiation, inflammation, and apoptosis [7].

The integrins are transmembrane αβ heterodimers, of which 18 α and eight β subunits are known in humans [8]. Integrins are broadly grouped into laminin-binding, collagen-binding, leukocyte and RGD-recognising types, and expression of the β1, β2 and β3 subfamilies is a constitutive activity of macrophages and monocytes. The fibronectin receptors α4β1 and α5β1 and the laminin receptor α6β1 are all key members of the β1 family, whereas the β2 subfamily is implicated in macrophage fusion, a hallmark of chronic inflammation. Among the β2 family members, integrin αMβ2 (Mac-1), is known to mediate cell–particle or cell–substrate interactions, although the complete role of Mac-1 in macrophage fusion leading to the formation of multinucleated giant cells remains unclear [9]. In the β3 family, αVβ3, a vitronectin receptor, is the most well described integrin.

Integrins on the surface of circulating leukocytes tend to be largely inactive [10] until inside-out or outside-in signaling triggers integrin-mediated adhesion. Inside-out signaling modifies how cells interact with their environment by facilitating receptor affinity and avidity to allow binding to extracellular ligands, while outside-in signaling mediates intracellular events in response to their environment by eliciting downstream signaling cascades in response to receptor occupation [11]. At the molecular level, macrophage adhesion to and activation by biomaterials, which ultimately results in the secretion of specific cytokines, could therefore be considered a complex mechanomolecular process involving dynamic and coordinated changes in integrin activation and/or binding, cytoskeletal reorganization, and subsequent intracellular signal transduction.

This plasticity of macrophage function allows them to be polarized into a spectrum of phenotypes, characterized by their profile of secreted cytokines, in response to foreign-body stimuli. As such, the M1 phenotype represents one end of this spectrum, characterized by the expression of high levels of proinflammatory cytokines, high production of reactive nitrogen and oxygen intermediates, promotion of a Th1 response, and strong microbicidal and tumoricidal activity. M0 macrophages, however, are defined as undifferentiated macrophages with the potential to be polarized into specific subtypes. Biomaterial-mediated modulation of the inflammatory response (particularly that by macrophages) is therefore currently the target of strategies aimed at enhancing tissue regeneration according to the principles of immunomodulation, which highlight the interaction between biomaterials and immune cells [1,2,4,12,13]. In the present study, we have examined the effect of titanium surface topography on the expression of macrophage integrin α1, α2, αM, β1 and β2 and concurrent cytokine secretion in both inflammatory (M1) and non-activated (M0) macrophages over the first 24 h following their attachment onto modified titanium surfaces.

Delineating the molecular events at the cell–biomaterial interface may facilitate the development of ‘tuneable’ biomaterials able to induce specific host immune responses that better promote tissue repair and regeneration.

## 2. Materials and Methods

### 2.1. Titanium

Titanium discs 3 mm in diameter × 1 mm thick were used in the study. A machined titanium surface acted as the control surface, and proprietary sandblasting and fluoridation processes (Dentsply Sirona Implants, Gothenburg, Sweden) were used to produce two rough surfaces: blasted and fluoride-modified. All discs were subsequently sealed and sterilized with γ-irradiation. The surface topography of the titanium discs was visualised using a high-vacuum scanning electron microscope (Jeol JCM-5000, Jeol Ltd., Tokyo, Japan), and profilometric analysis of the topographical characteristics of the titanium surfaces was performed using a LEXT OLS5000 Olympus laser confocal microscope (Olympus Australia Pty Ltd., Notting Hill, VIC, Australia). Profile roughness (Ra) and area roughness (Sa) parameters were quantified for each titanium surface.

### 2.2. Cell Culture

THP-1 monocytes were differentiated into a macrophage-like phenotype by incubation with 100 ng/mL of phorbol 12-myristate 13-acetate (PMA, Sigma-Aldrich, Castle Hill, NSW, Australia) for 24 h in complete media (RPMI 1640 supplemented with 10% foetal bovine serum and 50 units/mL of penicillin and 50 µg/mL streptomycin) at 37 °C in a 5% CO_2_ atmosphere. Differentiation of a non-inflammatory phenotype was achieved by removing the PMA-containing media after the initial 24 h stimulus, then incubating the cells in fresh complete media for a further 24 h. To induce an inflammatory macrophage phenotype, PMA-differentiated cells were incubated in complete media supplemented with 20 ng/mL lipopolysaccharide for 24 h. Adherent macrophages were detached using 0.25% trypsin prior to seeding onto the titanium discs in triplicate and incubated in complete media as required.

### 2.3. Macrophage Viability and Morphology

Macrophages seeded onto titanium discs (7.5 × 10^4^ cells/disc) were incubated for 24 h, after which viability was assessed. The discs were washed in PBS to remove nonadherent cells before incubation with 250µL of alamarBlue™ (Thermo Fisher Scientific, Seventeen Mile Rocks, QLD, Australia) diluted in complete media for 2 h. The fluorescence at 560/590 nm (excitation/emission) was subsequently measured using a spectrophotometer. To visually assess cell morphology following attachment and culture on the titanium surfaces, titanium discs with adherent macrophages were sputter-coated with gold and imaged using a scanning electron microscope (JCM-5000, Jeol, Tokyo, Japan). The diameter of the adherent cells was also quantitated using confocal microscopy imaging (Nikon A1, Tokyo, Japan) and subsequent image J analysis. The macrophages were fixed with 4% paraformaldehyde before staining the actin filaments with Phalloidin-California Red Conjugate (AAT Bioquest, Sunnyvale, CA, USA) and the cell nuclei with DAPI.

### 2.4. Cytokine Secretion

A multiplex ELISA (Bio-Plex Pro™ Human Th17 Cytokine Panel 15-Plex, Bio-Rad, Gladesville, NSW, Australia) was used to determine the concentrations of fifteen cytokines (IFN-γ, IL-1β, IL-4, IL-6, IL-10, IL-17A, IL-17F, IL-21, IL-22, IL-23, IL-25, IL-31, IL-33, sCD40L and TNF-α) secreted by M0 and M1 cells (7.5 × 10^4^ cells/disc) in the culture media after 24 h culture on the titanium discs.

### 2.5. Integrin Gene Expression

The expression of five integrin genes (α1, α2, αM, β1 and β2) in the M0 and M1 macrophages cultured on the three modified titanium discs (7.5 × 10^4^ cells/disc) was determined using quantitative real-time reverse transcription PCR after 24 h of in vitro culture. Total RNA was extracted from the seeded macrophage cells (RNeasy Mini Kit, QIAGEN, Germantown, MD, USA) and reverse-transcribed into cDNA using the SensiFAST™ cDNA Synthesis kit (Bioline, Aust Pty. Ltd., Auburn, NSW, Australia). The primer pairs used to amplify the target genes to allow the determination of relative expression are described in Table 1. The relative fold changes in gene expression were normalized to the housekeeping gene beta-Actin, and the expression levels of these genes in macrophages in response to the titanium discs were compared to expression levels following culture on tissue culture plastic.

### 2.6. Statistical Analysis

Data presented as means ± standard deviation (SD) of three replicate experiments were analysed using two-way analysis of variance (ANOVA), with post-hoc analysis corrected for multiple comparisons (Tukey) using GraphPad Prism (version 8). The confidence levels were taken to be 95% (*p* < 0.05).

## 3. Results

### 3.1. Titanium Surface Characterisation

Surface characterization was performed to determine the surface properties of the three different titanium discs. At low magnification (Figure 1A,D,G), concentric grooves from the milling process on an otherwise relatively smooth surface were observed on the machined titanium surface, while the blasted and fluoride-modified titanium surfaces were characterized by homogenous crater-like irregularities spread across the entire surface due to shot-peening the surface with TiO_2_ pellets. At higher magnifications (Figure 1B,E,H), the pronounced increase in surface features and the overall degree of roughness on the blasted and fluoride surfaces could be better appreciated. The fluoride-modified surface, initially prepared similarly to the blasted surface, was further acid-etched with hydrofluoric acid. At the highest magnification (Figure 1C,F,I), whilst similar to the blasted surface, nodular-like features could also be seen on the facets of the fluoride-modified surface.

Profilometry used to measure the vertical features on the surface of the titanium discs showed that the fluoride-modified discs had the highest profile roughness (Ra = 2.31 μm) and mean area roughness (Sa = 2.45 μm), followed by the blasted (Ra = 1.19 μm, Sa = 1.35 μm) and machined surfaces (Ra = 0.21 μm, Sa = 0.36 μm) (Table 2). Confocal imaging provided a further visual representation of the range of the surface height features (from pit to peak) for each titanium surface (Figure 2). Colour bars on the right side of the panels show the height range of vertical features for each analysed surface, with the fluoride-modified titanium having a much broader range (+22.255 to −27.794 µm) than the other two surfaces (blasted: +17.205 to −14.469 µm; machined: +9.031 to −4.037 µm) (Figure 2, Table 2).

### 3.2. Macrophage Viability and Morphology

The viability of M0 and M1 macrophages was similar on all three titanium surfaces (60–65%) after 24 h of culture. SEM analysis showed that the M0 cells displayed a rounded morphology with some filopodial extensions (Figure 3A). However, LPS-stimulated (M1) macrophages appeared larger, with extensive cell spreading and pseudopodia in close contact with the surface structures, particularly on the fluoride-modified titanium surface (Figure 3B), suggesting that this surface, at least in the short term (i.e., 24 h), may provide more favourable conditions for cell attachment. This was also supported by the significantly higher levels of cell viability (* *p* < 0.05) in both M0 and M1 cells cultured on the fluoride-modified titanium surface (Figure 3C).

Image J analysis used to quantitate the diameter of the M0 and M1 cells on the three titanium surfaces confirmed these observations; significant increases were shown in the cell diameters of M1 cells compared to M0 cells (* *p* < 0.0001) on all three surfaces. Moreover, culture of M1 cells on the fluoride-modified surface was shown to elicit the largest mean cell diameter compared to machined and blasted (^ *p* < 0.0001, Figure 3D).

### 3.3. Cytokine Analysis

In M0 macrophages, of the 15 cytokines assessed, only IL-1β, TNF-α and IL-31 were found to be secreted into the culture media at detectable levels following culture on the titanium surfaces (Figure 4). The low levels of these cytokines (mean < 2 pg/mL) in culture on tissue culture plastic (TCP) confirmed that these cells were in a non-inflammatory state. These M0 cells became more inflammatory after attachment to the fluoride-modified titanium surface for 24 h, when significant increases in IL-31, TNF-α and IL-1β levels were demonstrated (compared to culture on TCP). Significantly, increased mean levels of TNF-α from M0 cells were also found following culture of the M0 cells on the machined surface (^ *p* < 0.0001, Figure 4A).

In the M1 macrophages, comparison of mean levels of IL-1β (45.87 pg/mL), TNF-α (30.45 pg/mL) and IL-31 (39.45 pg/mL) secreted following culture on TCP with those for M0 cells (1.27, 5.07, and not detected) confirmed the proinflammatory status of these cells. Following attachment to the titanium surfaces for 24 h, IL-1β, TNF-α, IL-31, IL-23, sCD40L, IFN-γ, IL-22 and IL-25 were all shown to be secreted at detectable levels. Mean levels of IL-1β and IL-31 were significantly higher than that seen with M0 cells following culture on the machined surface (* *p* < 0.0001), while TNF-α secretion was significantly higher following M1 culture on the blasted and fluoride-modified surfaces (* *p* < 0.0001, Figure 5). Of the other secreted cytokines (IFN-γ, IL-22, IL-23, IL-25 and sCD40L), mean levels from M1 cells after 24 h culture on all three titanium surfaces were similar to levels with culture on TCP (Figure 4B).

### 3.4. Integrin Gene Expression

Integrin α1, α2, αM, β1 and β2 gene expression (Figure 5) was assessed at the same culture time point (24 h) to determine any potential correlation(s) with the concurrent cytokine secretion. Compared to expression on TCP, integrin α1 expression was downregulated (all expression fold changes <1) in M0 cells on all titanium surfaces (machined fold change 0.10 > blasted 0.22 > fluoride-modified 0.32 *p* < 0.001), whereas in M1 cells, α1 expression was relatively unchanged, i.e., fluoride-modified fold change = 1.29, blasted = 0.97 and machined = 0.78. Integrin α2 expression was upregulated in both M0 and M1 cells in response to the machined surface (7.4 and 5.1-fold, respectively). On the blasted and fluoride-modified surfaces, however, α2 expression was only upregulated in M1 cells (2.1 and 2.0-fold, respectively). Integrin β2 expression in M0 cells showed a similar pattern of expression to that seen for integrin α2 in response to the machined (2.37-fold), blasted (0.37-fold) and fluoride-modified surfaces (1.5-fold). In M1 cells, β2 expression was only upregulated on the fluoride-modified surface (1.81-fold) compared to the machined (0.73-fold) and blasted (0.76-fold) surfaces. In contrast, significant increases (*p* < 0.0001 compared to M0 cells) in both αM (machined 6.8-fold, blasted 2.0-fold, fluoride-modified 6.2-fold) and β1 expression (machined 22.3-fold, blasted 6.1-fold, fluoride-modified 5.9-fold) were observed in M1 cells on all three titanium surfaces.

## 4. Discussion

This study showed that macrophage integrin expression following culture of both non-inflammatory (M0) and inflammatory (M1) cells on titanium appears to be dependent upon both the titanium surface topography and cell phenotype. In general, in noninflammatory (M0) macrophages, only culture on the smoother (machined) titanium clearly upregulated the expression of integrins (α2, αM, β1 and β2) over the first 24 h. In M1-polarised inflammatory macrophages, however, expression of α2, αM and β1 integrins were all substantially upregulated over this same time period with culture on all three surfaces. Interestingly, M1 culture on the ‘rougher’ (blasted and fluoride-modified) titanium surfaces resulted in lower expression levels of integrins α2, αM and β1 compared to the ‘smoother’ machined surface. These differing surface-dependent patterns of integrin expression following culture on the three titanium surfaces over 24 h were also associated with differences in macrophage cytokine secretory responses, where IL-1β and IL-31 secretion was again significantly lower in M1 cells cultured on the rough (blasted and fluoride-modified) surfaces compared to the smooth (machined) surface. Whilst this does not prove a causal relationship between integrin expression and cytokine secretion, these results do support the hypothesis that specific integrin expression due to titanium surface modulation could subsequently modulate later cytokine secretion. While this study suggests that differences in surface roughness may be a significant factor influencing macrophage adhesion and integrin expression, other potential differences between the test materials resulting from manufacturing processes may also play a role in these results.

Whether these differences in integrin expression in response to adhesion to the different titanium surfaces and subsequent cytokine secretion profile can modulate a biomaterial-mediated inflammatory response remains to be further examined in vivo. In vitro studies using human monocytic cell lines such as THP-1, U937, MonoMac 6, ML-2 and HL-60 are frequently used to study monocyte/macrophage differentiation and function [14]. The THP-1 used in this study are highly plastic and sensitive to many stimuli, and therefore can be polarized into multiple lineages [15] (for review). PMA treatment of THP-1 cells leads to a more mature phenotype with a lower rate of proliferation, higher levels of adherence, higher rate of phagocytosis and increased cell-surface expression of CD11b and CD14, although this differentiation phenotype has been shown to be variable between researchers [16]. PMA-differentiated THP-1 macrophages, however, do not entirely reproduce the response spectrum of primary-monocyte-derived macrophages to activating stimuli. Despite these differences, it is generally accepted that THP-1 should be regarded as a simplified model of human macrophages when investigating relatively straightforward biological processes such as polarization and its functional implications [17].

In using these cells (THP-1), it is also important to note that studies have shown that the methodology used to detach adherent macrophages in culture for subsequent seeding experiments, including trypsin (0.25% used in this study), can significantly alter the subsequent inflammatory response [18,19] and consequently the observed secretory cytokine profile of biomaterial-adherent macrophages. However, our results suggest that the transfer procedures used to seed the titanium discs had little effect on macrophage phenotype, as IL-1β, TNF-α and IL-31 levels in M0 cells all remained very low when seeded onto TCP over 24 h of culture. Conversely, in the M1 cells, elevated cytokine (IL-1β, TNF-α, IL-31) levels seen on TCP were stimulated further by culture on titanium.

It was necessary to describe the surfaces under examination, as the modified titanium surfaces used in the study may each contain more than one novel surface characteristic, e.g., differing surface topography and or surface chemistry [20,21,22]. While trace amounts of fluoride ions are known to primarily affect osteoprogenitor cells and undifferentiated osteoblasts to enhance bone formation, the potential immunomodulatory effects of fluoride ions on macrophages are still unclear. Some studies have shown concentration-dependent effects, whereby ultralow concentrations of fluoride ions activated RAW 264.7 macrophages with increased expression of proinflammatory genes (IL-6 and IL-1β), while micromolar (2.4–24 µM) concentrations downregulated expression of M1 markers (iNOS) and upregulated M2 markers (ARG), suggesting that fluoride may be an effective osteoimmunomodulatory agent [23]. In contrast, other studies have shown that micromolar fluoride concentrations reduced the macrophage population and significantly increased the secretion of proinflammatory cytokines [24,25]. More recently, a nanostructured TiF*_X_*/TiO*_X_*-coated titanium surface was shown to stimulate adherent macrophage elongation and elicit favourable osteoimmunomodulation by 72 h [26].

Morphological changes induced by titanium surface modification are known to be related to a cell’s differentiation and functional status, i.e., polarized or stretched cells are considered to be activated, whereas stable or steady oval macrophages are considered inactivated [27,28]. Indeed, M1 macrophages were demonstrated to have a greater mean cell diameter (compared to M0) on all three titanium surfaces, where cell diameters on fluoride-modified and machined surfaces were greater than that seen on the blasted surfaces. This suggested higher inflammatory activation by the fluoride-modified and machined surfaces, which was subsequently shown to correlate with higher mean levels of IL-1β and IL-31 on these surfaces compared to levels from cells cultured on the blasted surface.

The surface roughness of the biomaterial has also been shown to influence protein adsorption, with more proteins adsorbed onto rougher surfaces [29]. While changes in surface topography were indeed evident in the titanium discs used in this study, differences in the macrophage response(s) described in the results could also reflect differences in the surface chemistry of the test materials, as cells have a limited capacity to adhere directly to a non-proteinaceous material [30]. Integrin binding sites are provided by the coagulation proteins, platelets, complement, and other soluble serum and blood proteins that adsorb onto the surface of implanted biomaterials within seconds after implantation. Differential protein adsorption onto the surface due to changes in surface chemistry and/or wettability following surface treatment [31] may therefore affect subsequent integrin expression in cells following attachment. We have previously shown in vivo, by proteomic analysis of both titanium surface adherent and wound exudate material, that during early osseous healing, titanium surface hydrophilicity promoted an immunomodulatory pro-reparative environment [32]. While the hydrophilicity / wettability of the surfaces was not assessed in this study, published data does suggest that the test surfaces in this instance were all similarly hydrophobic [33]. Unfortunately, any further proteomic analysis of adsorbed surface proteins is beyond the scope of this study.

Cell adhesion plays an integral role in enabling communication between cells and their microenvironment, and integrins are well known to influence inflammation and macrophage polarization. Integrin α2β1, for example, has been shown to have a direct effect on macrophage phenotype [34]. Interestingly, the significant decreases in α2 and β1 gene expression in M1 cells on blasted and fluoride-modified surfaces seen in the present study were associated with significantly lower secretion of IL-1β and IL-31 from these cells, suggesting that these surfaces may promote immunomodulation, resulting in a less inflammatory phenotype, as has been demonstrated with macrophage culture on other topographically modified titanium surfaces [4].

Within the limitations of the study, these results showed that integrin expression and cytokine secretion in macrophages following their attachment to surface-modified titanium was dependent upon macrophage phenotype. This suggests that modulation of macrophage inflammatory response to titanium implantation via the differential expression of integrins may be biologically plausible, and hence a promising tool for ameliorating adverse immune reactions and accelerating implant integration.

## Figures and Tables

**Figure 1 jfb-14-00211-f001:**
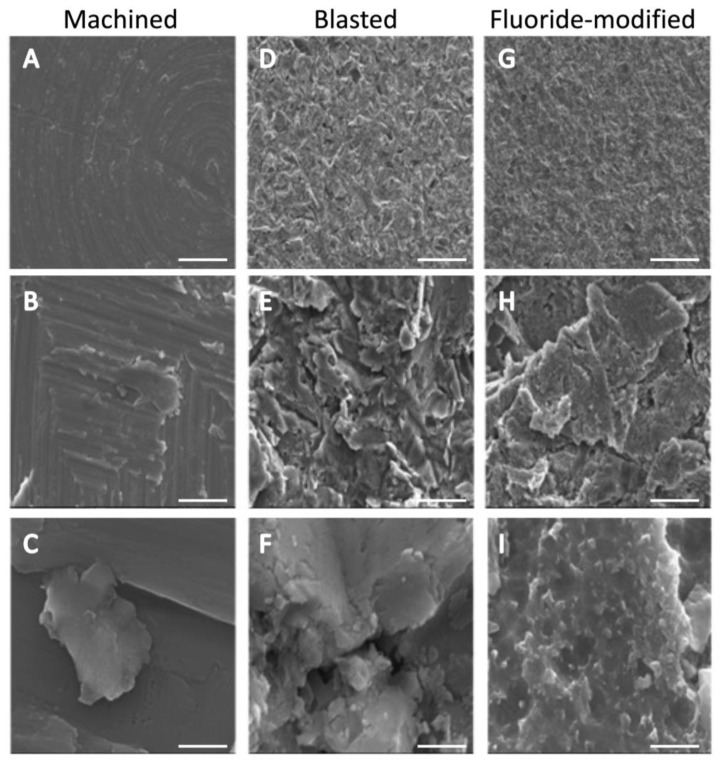
Titanium surface analysis: SEM images at increasing magnification of the three titanium surfaces (Machined (**A**–**C**), Blasted (**D**–**F**) and Fluoride-modified (**G**–**I**)). White scale bar = 100 µm (**top panel**), 10 µm (**middle panel**), and 1 µm (**bottom panel**), respectively.

**Figure 2 jfb-14-00211-f002:**
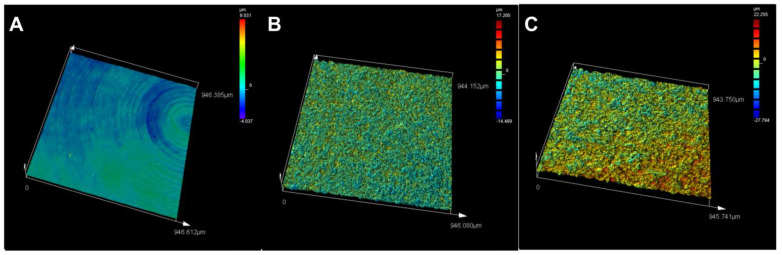
Confocal images of the titanium surfaces ((**A**) machined, (**B**) blasted and (**C**) fluoride-modified) illustrate the range in surface roughness, i.e., smoother (blue–green) to rougher (yellow–red). Quantitative analysis of the surface height differences (scale bar top right) shows fluoride-modified (+22.255 to −27.794 µm) > blasted (+17.205 to −14.469 µm) > machined (+9.031 to −4.0 µm).

**Figure 3 jfb-14-00211-f003:**
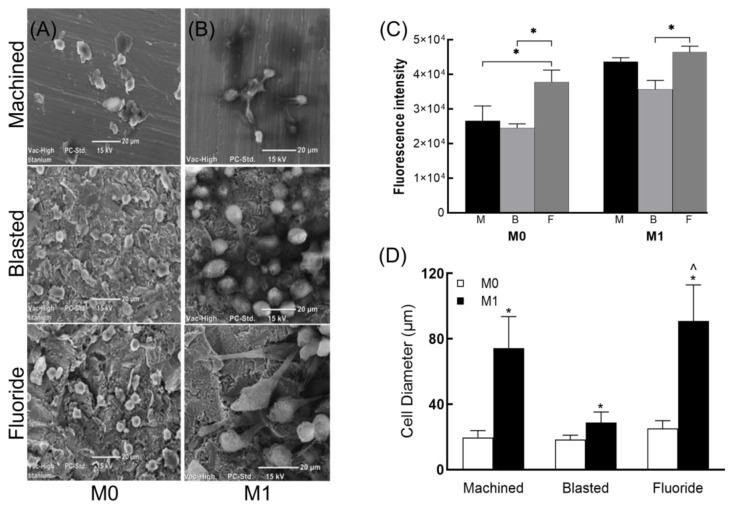
Macrophage morphology: representative confocal microscopy images of (**A**) M0 and (**B**) M1 cells on machined, blasted and fluoride modified titanium discs 24 h post surface seeding suggests greater cell spreading in activated (M1) cells. (**C**) Cell viability of both M0 and M1 cells was higher in cells cultured on the fluoride-modified titanium discs (* *p* < 0.05). (**D**) The mean diameter of M1 cells (compared to M0 cells) was significantly increased (* *p* < 0.0001) on all three surfaces. Moreover, culture of M1 cells on the fluoride-modified surface (compared to machined and blasted) was shown to elicit the largest mean cell diameter (^ *p* < 0.0001).

**Figure 4 jfb-14-00211-f004:**
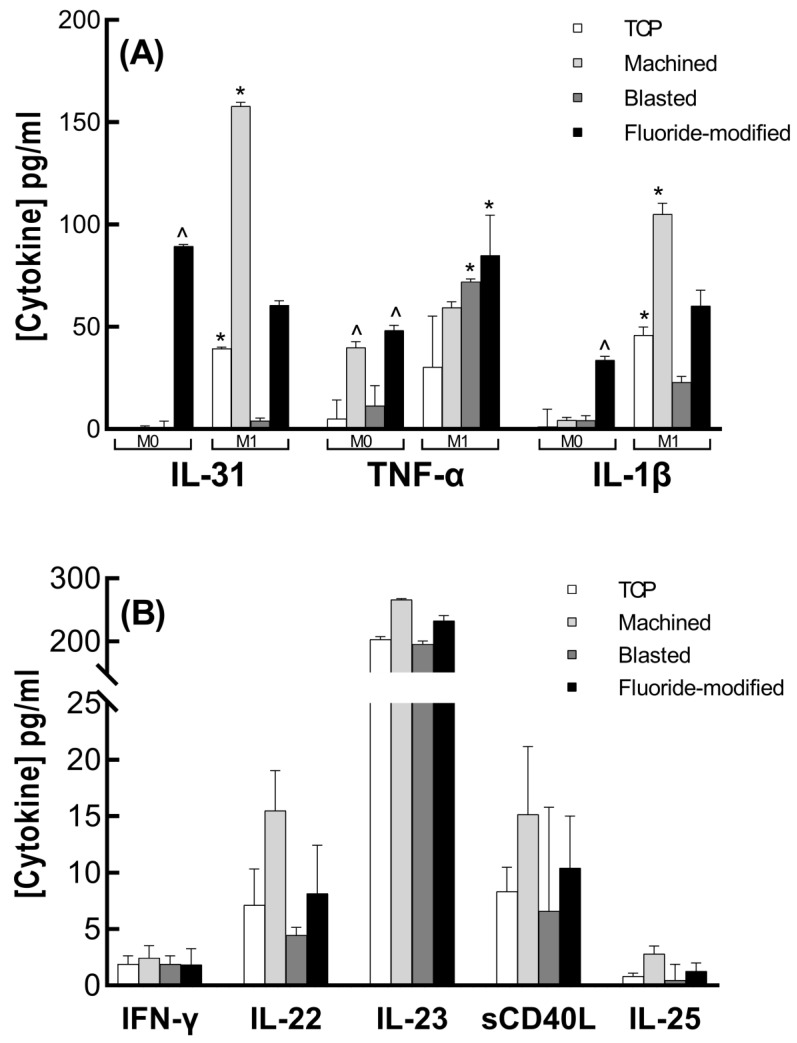
Histograms show (**A**) the mean (±SD for 3 replicates) concentration (pg/mL) of cytokines secreted by both M0 and M1 macrophages when cultured on tissue culture plastic (TCP) and the three titanium surfaces (machined, blasted and fluoride-modified) over 24 h. ^ denotes a significant difference compared to TCP (*p* < 0.0001). * denotes a significant difference in M1 levels compared to M0 levels (*p* < 0.001). (**B**) Mean (±SD for 3 replicates) concentration (pg/mL) of cytokines secreted by the M1 macrophages only when cultured on TCP and the three titanium surfaces for more than 24 h.

**Figure 5 jfb-14-00211-f005:**
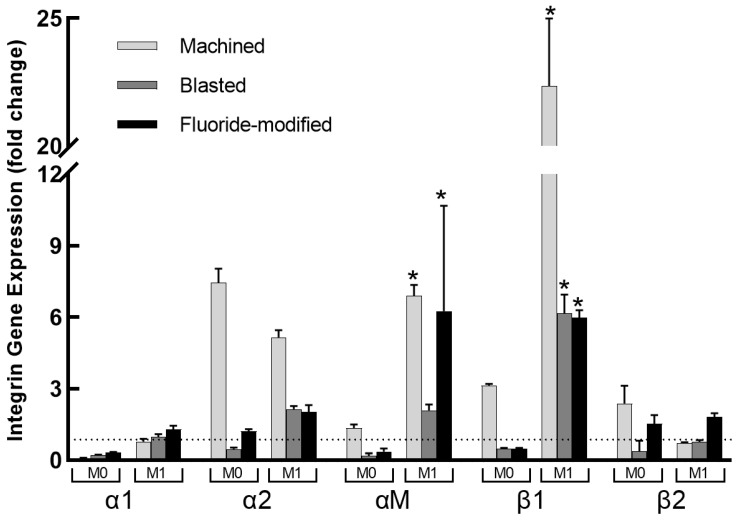
Integrin α1, α2, αM, β1 and β2 fold change expression (vs. TCP) in M0 and M1 macrophages cultured on machined (M), blasted (B) and fluoride-modified (F) titanium surfaces. Means ± SDs shown for 3 replicates. * denotes significant difference (*p* < 0.0001) in fold change (M1 compared to M0). …. denotes fold change = 1.

**Table 1 jfb-14-00211-t001:** qPCR primer pairs.

Gene	Forward Primer	Reverse Primer
ITGA1	GGTGCCCGAAGAGGAGTTAAAA	TCCTCGGTTATAGCTGCCAAGA
ITGA2	TCAGGGCACTATCCGCACAAAGTA	CCAAAGGCACCAATAGACACATCG
ITGAM	TGATGCTGTTCTCTACGGGGAGCA	AACAGGTAAACAGCACCCCGGTTG
ITGB1	TGAGCTGGACAGAGGAGGAGGAAG	GCCTCCTGCTGCTCAATGATGC
ITGB2	GAAGGAAGCTGCCGGAAGGACAAC	GCGCTCACAGTTGATGGTGTCACA
ACTB	CACCATTGGCAATGAGCGGTTC	AGGTCTTTGCGGATGTCCACGT

**Table 2 jfb-14-00211-t002:** Surface roughness parameters. Ra: arithmetic average profile roughness; Rz: mean maximum height; Sa: mean area roughness; Height Profile: mean height profile range.

	Ra (µm)	Rz (µm)	Sa (µm)	Height Profile (µm)
Machined	0.214	1.803	0.361	+9.031 to −4.037
Blasted	1.196	10.146	1.351	+17.205 to −14.469
Fluoride-modified	2.31	20.226	2.449	+22.255 to −27.794

## Data Availability

Data are available on request from the corresponding author.

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
