# Peer review of "The Effect of Titanium Surface Topography on Adherent Macrophage Integrin and Cytokine Expression"

_jfb, 2023, doi:10.3390/jfb14040211_

Round 1

Reviewer 1 Report

This is a review report for JFB-287757. This paper show differences in integrin expression and cytokine elaboration between macrophages cultured on different clinically-relevant implant surfaces. This is a timely paper that is well-executed and well-presented. It was a pleasure to read. I have a few comments below to help the authors but I look forward to seeing this eventually published. 

The authors should mention that macrophage polarization exists on a spectrum and that M0, M1, etc. is useful but not fully descriptive of some nuisance.

I am curious if there are other papers that show a more thorough characterization of Blasted and Fluoride-modified surfaces – adding those reference would be useful.

Do the authors have any further speculation on the effect of Fluoride on macrophages beyond what they included? I think a little more speculation is appropriate.

The authors should be more explicit about sample sizes.

Sometimes it appears as IFNg instead of IFNγ; please fix.

The discussion was highly informative, thank you. 

Author Response

The authors should mention that macrophage polarization exists on a spectrum and that M0, M1, etc. is useful but not fully descriptive of some nuisance.

Response: We thank the reviewer for their review of the manuscript. As suggested a comment on macrophage phenotype has been included in the introduction lines 78 – 85.

“This plasticity of macrophage function, allows them to polarize into a spectrum of phenotypes, characterized by their profile of secreted cytokines, in response to foreign-body stimuli. As such, the ‘M1’ phenotype represents one end of this spectrum characterized by the expression of high levels of proinflammatory cytokines, high production of reactive nitrogen and oxygen intermediates, promotion of a Th1 response, and strong microbicidal and tumoricidal activity. ‘M0’ macrophages however are defined as undifferentiated macrophages with the potential to polarize into specific subtypes.”

I am curious if there are other papers that show a more thorough characterization of Blasted and Fluoride-modified surfaces – adding those reference would be useful.

Response: The test materials were the same as those used clinically and provided by the manufacturer. As such, they have been well characterised in the literature and additional references [20-22] have been included (line 307).

[20] Masaki, C., Schneider, G.B., Zaharias, R., Seabold, D., Stanford, C. Effects of implant surface microtopography on osteoblast gene expression. Clin Oral Implants Res. 2005, 16, 650 – 656. doi: 10.1111/j.1600-0501.2005.01170.x

[21] Monjo, M., Petzold, C., Ramis, J.M., Lyngstadaas, S.P., Ellingsen, J.E. In vitro osteogenic properties of two dental implant surfaces. Int J Biomater. 2012, 181024. doi: 10.1155/2012/181024.

[22] Kang, B.S., Su, Y.L., Oh, S.J., Lee, H.J., Albrektsson, T. XPS, AES and SEM analysis of recent dental implants. Acta Biomaterialia. 2009, 5, 2222 – 2229. doi: 10.1016/j.actbio.2009.01.049.

Do the authors have any further speculation on the effect of Fluoride on macrophages beyond what they included? I think a little more speculation is appropriate.

Response: Further discussion added (lines 310 – 319) and references updated.

“Some studies have shown concentration dependent effects, whereby ultralow concentrations of fluoride ions activated RAW 264.7 macrophages with increased expression of pro-inflammatory genes (IL6 and IL1β), while micromolar (2.4 - 24µM) concentrations downregulated M1 marker (iNOS) and upregulated M2 marker (ARG) expression, suggesting fluoride may be an effective osteoimmunomodulatory agent [23]. In contrast, other studies have shown micromolar fluoride concentrations reduced the macrophage population and significantly increased the secretion of pro-inflammatory cytokines [24, 25]. More recently, a nanostructured TiFX/TiOX coated titanium surface was shown to stimulate adherent macrophage elongation and elicited favourable osteoimmunomodulation by 72 h [26].”

The authors should be more explicit about sample sizes.

Response: Sample size added to line 147 “Data presented as means ± standard deviation (SD) of three replicate experiments….”

Sometimes it appears as IFNg instead of IFNγ; please fix.

Response: Manuscript checked for uniformity.

Reviewer 2 Report

In the manuscript " The Effect of Titanium Surface Topography on Adherent Macrophage Integrin and Cytokine Expression", Pitchai et al. fabricated and characterized three types of titanium surface to examine the effect of modified surface on the macrophage adhesion and cytokine secretion. The macrophage integrin expression and cytokine secretion of Non-polarised (M0) and inflammatory (M1) polarised macrophage were investigated in vitro. It was demonstrated that adherent inflammatory macrophages can interact with titanium in a surface dependent manner. The work is scientifically interesting and well written. However, there are several concerns concerning implementation of the approach.

1.     It is considerable that the two rough surface is by sandblasting and fluoride process, which should definitely change the surface composition of the titanium surface. But in the section of material characterization, certain characterization method such as EDS, XPS, XRD and topological parameter such as wettability test etc. is missing. It seemed that the article set surface roughness as a single influence factor of macrophage adhesion, which is not that convincing.

2.         Regarding the macrophage morphology, macrophage appeared larger with extensive cell spreading and pseudopodia, and have significantly higher level of cell viability particularly on the Fluoride-modified titanium surface. It is well known that fluoride exposure could lead to increased ROS production and enhanced inflammatory reaction. It is reasonable to doubt that the fluorine on the fluoride modified titanium surface can have an effect on macrophage adhesion and proliferation. Authors should try to at less explain, or even eliminate the possible influence of this factor.

3.     Concerning cytokine analysis, M0 cells became more inflammatory after attachment to the Fluoride- modified titanium surface for 24h, and M1 cells have a significantly higher TNF-α secretion on Fluoride- modified titanium surface. It was suggested that fluoride can increase concentration of PGE2 and TXA2 in THP1 macrophages through activating the transcription factor, nuclear factor-kappa B that initiate and develop the inflammatory process. ( Toxicol In Vitro. 2015 Oct;29(7):1661-8. doi: 10.1016/j.tiv.2015.06.024.) This could be the main reason that macrophage cells appears to be more inflammatory on fluoride-modified titanium surface.

Author Response

It is considerable that the two rough surface is by sandblasting and fluoride process, which should definitely change the surface composition of the titanium surface. But in the section of material characterization, certain characterization method such as EDS, XPS, XRD and topological parameter such as wettability test etc. is missing. It seemed that the article set surface roughness as a single influence factor of macrophage adhesion, which is not that convincing.

Response: We thank the reviewer for their comments. We certainly agree other factors may indeed affect the macrophage adhesion response, however given the ample literature on the effects of surface topography on macrophage phenotype we feel it’s not unreasonable to highlight this difference as a possible explanation of the results observed. Nonetheless we have added this point in the manuscript (lines 275 -279).

“While this study suggests differences in surface roughness may be a significant factor influencing macrophage adhesion and integrin expression, other potential differences between the test materials (but not directly examined) resulting from the manufacturing processes may also play a role in these results.”

Regarding the macrophage morphology, macrophage appeared larger with extensive cell spreading and pseudopodia, and have significantly higher level of cell viability particularly on the Fluoride-modified titanium surface. It is well known that fluoride exposure could lead to increased ROS production and enhanced inflammatory reaction. It is reasonable to doubt that the fluorine on the fluoride modified titanium surface can have an effect on macrophage adhesion and proliferation. Authors should try to at less explain, or even eliminate the possible influence of this factor.

Response: Further discussion on fluoride added lines 310 – 319.

“Some studies have shown concentration dependent effects, whereby ultralow concentrations of fluoride ions activated RAW 264.7 macrophages with increased expression of pro-inflammatory genes (IL6 and IL1β), while micromolar (2.4 - 24µM) concentrations downregulated M1 marker (iNOS) and upregulated M2 marker (ARG) expression, suggesting fluoride may be an effective osteoimmunomodulatory agent [23]. In contrast, other studies have shown micromolar fluoride concentrations reduced the macrophage population and significantly increased the secretion of pro-inflammatory cytokines [24, 25]. More recently, a nanostructured TiFX/TiOX coated titanium surface was shown to stimulate adherent macrophage elongation and elicited favourable osteoimmunomodulation by 72 h [26].”

Concerning cytokine analysis, M0 cells became more inflammatory after attachment to the Fluoride- modified titanium surface for 24h, and M1 cells have a significantly higher TNF-α secretion on Fluoride- modified titanium surface. It was suggested that fluoride can increase concentration of PGE2 and TXA2 in THP1 macrophages through activating the transcription factor, nuclear factor-kappa B that initiate and develop the inflammatory process. ( Toxicol In Vitro. 2015 Oct;29(7):1661-8. doi: 10.1016/j.tiv.2015.06.024.) This could be the main reason that macrophage cells appears to be more inflammatory on fluoride-modified titanium surface.

Response: The authors agree and have added the reference [25] at line 317.

Round 2

Reviewer 2 Report

I recommend the publication of this manuscript.

Author Response

Thank you for your comments